# Comprehensive Characterization of a *Streptococcus agalactiae* Phage Isolated from a Tilapia Farm in Selangor, Malaysia, and Its Potential for Phage Therapy

**DOI:** 10.3390/ph16050698

**Published:** 2023-05-05

**Authors:** Megat Hamzah Megat Mazhar Khair, An Nie Tee, Nurul Fazlin Wahab, Siti Sarah Othman, Yong Meng Goh, Mas Jaffri Masarudin, Chou Min Chong, Lionel Lian Aun In, Han Ming Gan, Adelene Ai-Lian Song

**Affiliations:** 1Department of Microbiology, Faculty of Biotechnology and Biomolecular Sciences, Universiti Putra Malaysia, Serdang 43400, Selangor, Malaysia; gs63305@student.upm.edu.my (M.H.M.M.K.);; 2Department of Cell and Molecular Biology, Faculty of Biotechnology and Biomolecular Sciences, Universiti Putra Malaysia, Serdang 43400, Selangor, Malaysiamasjaffri@upm.edu.my (M.J.M.); 3Institute of Bioscience, Universiti Putra Malaysia, Serdang 43400, Selangor, Malaysia; 4Department of Veterinary Preclinical Sciences, Faculty of Veterinary Medicine, Universiti Putra Malaysia, Serdang 43400, Selangor, Malaysia; 5Nanomaterials Synthesis and Characterisation Laboratory, Institute of Nanoscience and Nanotechnology, Universiti Putra Malaysia, Serdang 43400, Selangor, Malaysia; 6Department of Aquaculture, Faculty of Agriculture, Universiti Putra Malaysia, Serdang 43400, Selangor, Malaysia; 7Department of Biotechnology, Faculty of Applied Sciences, UCSI University, Kuala Lumpur 56000, Selangor, Malaysia; lionelin@ucsiuniversity.edu.my; 8Patriot Biotech, Sunway Geo Avenue, Bandar Sunway, Subang Jaya 47500, Selangor, Malaysia

**Keywords:** *Streptococcus*, phage, endolysin, aquaculture, antimicrobial

## Abstract

The *Streptococcus agalactiae* outbreak in tilapia has caused huge losses in the aquaculture industry worldwide. In Malaysia, several studies have reported the isolation of *S. agalactiae*, but no study has reported the isolation of *S. agalactiae* phages from tilapia or from the culture pond. Here, the isolation of the *S. agalactiae* phage from infected tilapia is reported and it is named as vB_Sags-UPM1. Transmission electron micrograph (TEM) revealed that this phage showed characteristics of a *Siphoviridae* and it was able to kill two local *S. agalactiae* isolates, which were *S. agalactiae* smyh01 and smyh02. Whole genome sequencing (WGS) of the phage DNA showed that it contained 42,999 base pairs with 36.80% GC content. Bioinformatics analysis predicted that this phage shared an identity with the *S. agalactiae* S73 chromosome as well as several other strains of *S. agalactiae*, presumably due to prophages carried by these hosts, and it encodes integrase, which suggests that it was a temperate phage. The endolysin of vB_Sags-UPM1 termed Lys60 showed killing activity on both *S. agalactiae* strains with varying efficacy. The discovery of the *S. agalactiae* temperate phage and its antimicrobial genes could open a new window for the development of antimicrobials to treat *S. agalactiae* infection.

## 1. Introduction

*Streptococcus agalactiae*, or Lancefield’s Groups B streptococcus (GBS), is a Gram-positive coccus that is beta-hemolytic and a facultative anaerobe. It is divided into ten molecular serotypes (serotype Ia, Ib, II–IX) based on the capsular polysaccharide (cps) gene and can be further classified based on clonal complex (CC), sequence type (ST), phage type, and virulence factor [1,2,3,4]. Its invasion into neonates during pregnancies is one of the leading causes of newborn sepsis and meningitis [3]. It also causes streptococcosis in tilapia, which results in significant mortality, leading to economic loss to the industry [5]. The widespread outbreak of *S. agalactiae* in fish farms in Southeast Asian countries such as Malaysia, Singapore, Thailand, and Vietnam is caused predominantly by serotype III, ST283 strains, and it was recently implicated with food-borne infection in healthy human adults [6,7,8]. The high genetic similarity between fish isolates from Thailand and human isolates from Singapore suggested that *S. agalactiae* transmission from fish to humans might have occurred at an alarming rate [7]. The tendencies of farmers to use antibiotics to control streptococcal outbreak also raises concerns regarding the transmission of antimicrobial resistant (AMR) bacteria to humans, which further complicates the treatment of multidrug-resistant *S. agalactiae* and other bacteria in fish and humans [9].

Phages are bacteria killing agents that have shown potential in antimicrobial therapy. Phages are divided into two types, lytic and temperate phages, depending on the expression of genes such as integrase that allows the incorporation of phages into the host chromosome for the latter. Lytic phages are preferred for phage therapy due to their ability to ultimately lyse the host, whereas temperate phages could become dormant in the host, which prevents lysis of the host. There have been a number of reports on *S. agalactiae* phages isolated from cow mastitis [10,11], wastewater plants [12], and mitomycin-C-induced culture of *S. agalactiae* clinical strains from vaginal or neonatal infection [13,14]. However, reports on phage isolation in streptococcal-infected aquaculture are far fewer, with only a few reported thus far: *S. agalactiae* phage HN48 from tilapia (*Oreochromis niloticus*) [15], *Streptococcus iniae* phage from olive flounder (*Paralichthys olivaceus*) [16], and *Streptococcus parauberis* phage Str-PAP-1 from olive flounder [17]. It is notable that all *S. agalactiae* phages isolated thus far are temperate phages, which are not preferred for phage therapy [12]. Only recently, four lytic phages were isolated from aquaculture farms which had specificity towards *S. agalactiae* serotype Ia [18]. Other than whole phages, a phage-encoded lytic enzyme called endolysin isolated from *S. agalactiae* phage and prophage has also been reported to show bactericidal effects in several streptococcal hosts, and thus can be further explored for antimicrobial therapy. Previously, B30 and PlyGBS endolysins were isolated from *S. agalactiae* prophages and exhibited killing activity in groups A, B, C, and G streptococci [14,19]. In addition, LambdaSa01 and LambdaSa02 endolysins, which were isolated from the *S. agalactiae* prophage, were active against *S. agalactiae*, *Streptococcus pneumoniae*, and *Staphylococcus aureus* [20]. However, none of these phages were isolated from the aquaculture environment, but rather from clinical samples, and no reports have tested the application of phage endolysins in aquaculture settings.

The purpose of the current study was to explore new alternatives in treating *S. agalactiae* infection in fish by using phages or phage-encoded endolysins. To achieve this, a novel phage against *S. agalactiae*, named vB_Sags-UPM1, was isolated from deceased tilapia from a farm that had an outbreak. The characteristics of vB_Sags-UPM1, such as morphology, growth kinetics, lysogeny efficiency, and host range, were determined. The whole genome of the phages was sequenced and coding sequences were annotated using bioinformatics tools to compare its origin and to isolate the putative endolysin gene. Subsequently, the putative endolysin gene, which was named Lys60, was cloned into *Escherichia coli*, expressed, and tested to determine its killing capacity on local *S. agalactiae* isolates. Therefore, this study aims to explore new antimicrobial alternatives to treat *S. agalactiae* infection, specifically in tilapia.

## 2. Results

### 2.1. Characterization of vB_Sags-UPM1 in S. agalactiae smyh01 and smyh02

*Streptococcus agalactiae* phage vB_Sags-UPM1 produced halo zones in *S. agalactiae* smyh01 and smyh02, and no halo zones in *S. pyogenes* and *S. aureus* PS88 when a high-titer phage (10^9^ PFU/mL) was spotted on the lawns of the bacteria (Appendix A). It formed small (0.6 ± 0.2 mm in diameter) and turbid plaques upon plating on BHI double-layer agar seeded with *S. agalactiae* smyh01 (Figure 1a). The phage particle under TEM showed morphology of an icosahedral head (50 ± 10 nm) and non-contractile tail (110 ± 25 nm), which was suggestive of a *Siphoviridae* family (Figure 1b). The one-step growth curve of the phage in *S. agalactiae* smyh01 showed a latent period of 30 min and a burst size of 356.53 ± 38.53 PFU/cell, whereas the latent period and burst size of the phage in *S. agalactiae* smyh02 was 40 min and 99.48 ± 14.44 PFU/cell, respectively (Figure 2a). There was a significant difference detected when comparing the burst size of the phage in both hosts (*p* = 0.0004). For vB_Sags-UPM1 killing activity, there was a significant difference of OD_600nm_ reading between the control and infected bacteria for both hosts (*p* < 0.00010) from the time of virus inoculation (0 h post-inoculation) until 24 h post-inoculation (Figure 2b). The lysogeny efficiency of smyh01 and smyh02 was 9.02% ± 4.35 and 40.86% ± 12.79, respectively, with a significant difference between the two strains (*p* = 0.015).

### 2.2. Bioinformatics

#### 2.2.1. Overall Genome of vB_Sags-UPM1

The genome of vB_Sags-UPM1 was a negative strand DNA with a size of 42,999 bp and GC content of 36.80%. Upon query using BLASTn, the phage genome shared the highest homology with the prophage of *S. agalactiae* strain S73 (Accession: CP030845), with an ANI of 96.33% and nucleotide coverage of 1.51%. In the virus taxonomy database, the vB_Sags-UPM1 genome shared the most homology with phage Javan53 (Accession: MK448796), a prophage of *S. agalactiae* strain ILRI005 [21] with an ANI of 88.77% and nucleotide coverage of 47.27%. Gene prediction resulted in 65 ORFs that constituted modules typically found in tailed phages such as morphogenesis, packaging, lysis, replication, regulatory, and lysogeny (Table 1). The modular organization was similar to previously sequenced genomes of temperate streptococcal *Siphoviridae*, which was arranged from DNA replication/regulatory-packaging-morphogenesis-lysis-lysogeny [22]. The DNA replication/regulatory module consisted of nucleases, DNA polymerase/primase, helicase, DNA methylase, helix-turn-helix (HTH) transcriptional regulators, and DNA binding proteins. The packaging module contained terminase and portal proteins. The morphogenesis module consisted of genes that made up the *Siphoviridae* structure, such as the tail, head–tail connector, head, and capsid proteins. The lysis module was comprised of glycoside hydrolase-25 (GH25) lysozyme and holin proteins. The lysogeny module contained the integrase gene, and it was situated at the positive strand in between lysis and replication modules. Figure 3a summarizes the whole genome map of the vB_Sags-UPM1 genome. The genome of vB_Sags-UPM1 shared a close homology with prophages of *S. agalactiae* S73, SG-M29 (Accession: CP021866), SG-M158 (Accession: CP021864), SG-M1 (Accession: CP012419), SG-M50 (Accession: CP021865), SG-M163 (Accession: CP021863), SGEHI2015-95 (Accession: CP025028), and FWL1402 (Accession: CP016391) strains, while being distantly related to prophages of *S. pyogenes* such as NCTC8320 (Accession: LS483391) strain (Figure 3b).

The pairwise comparison of the vB_Sags-UPM1 genome and the genome of *S. agalactiae* S73 prophage revealed homology in packaging, morphogenesis, lysis, and lysogeny modules, whereas DNA replication/regulatory module was poorly conserved (Figure 4a). When comparing the vB_Sags-UPM1 genome to the genomes of *S. agalactiae* SG-M29, SG-M158, and SGEHI2015-95 prophages, homology was observed in DNA replication/regulatory, lysogeny, lysis, and tail morphogenesis module; however, the lysis module of SGEHI2015-95 was not located in the prophage sequence in the host genome (Figure 4b–d). For *S. agalactiae* prophages SGEHI2015-25 and A909 (LambdaSa03), sequence homology with vB_Sags-UPM1 genome was found in a small region of tail morphogenesis and lysis modules; however, SGEHI2015-25 also shared a similar lysogeny module with vB_Sags-UPM1 (Figure 4e,f). Although the A909 strain possessed two additional prophage sequences in its genome (LambdaSa04 and LambdaSa05), there was no homology detected between vB_Sags-UPM1 with these prophages. The comparison of the vB_Sags-UPM1 genome with its closest relative in the virus taxonomy, Javan53 phage, showed homology only in the lysis module (data not shown). Overall, the DNA pairwise comparison among several prophages and vB_Sags-UPM1 demonstrated genetic mosaicism, and highly conserved regions were observed in the lysogeny, lysis, and the end region of the morphogenesis module, which encoded a putative tail tip protein that contained glucosaminidase (Figure 5a). The packaging and morphogenesis module of vB_Sags-UPM1 might arise from S73, whereas the DNA replication/regulatory module might arise from SG-M29 or other similar prophage sequences (Figure 5b).

#### 2.2.2. DNA Replication/Regulatory Module

The genes encoding for helicase, helix-turn-helix (HTH) transcriptional regulators, phage antirepressor KilAC domain-containing protein, phage resistance protein, bifunctional DNA primase/polymerase, VRR-NUC domain-containing protein, ArpU family transcriptional regulator, and ImmA/IrrE family metallo-endopeptidase in vB_Sags-UPM1 were more than 99% identical to those in SG-M29, SG-M158, and SGEHI2015-95 prophage sequences (Figure 4b–d). HTH is a key DNA-binding domain found in cro/cI repressors that controls the switching between the lytic and lysogenic life cycle in lambda phages by binding to their opposing promoters [23]. Cro and cI repressors bind to PRM and PR promoters, respectively, downregulating the expression of lysogenic genes (cro-bound PRM) and lytic genes (cI-bound PR) [23]. The cro/cI repressor was predicted in the LambdaSa03 sequence but only shared 30–70% homology with HTH transcriptional regulators of vB_Sags-UPM1. The KilAC domain was found to be part of a regulatory protein ArisA, together with the AntA/B domain [24]. It coordinated the expression of lytic genes under SOS condition in Listeria monocytogenes that contained two prophages in its genome [24]. VRR-NUC domain-containing protein, ArpU family transcriptional regulator, and ImmA/IrrE family metallo-endopeptidase were involved in the type III restriction-modification system, the regulation of cellular muramidase-2, and the conjugative transposon transfer, respectively, but their roles in streptococcal phages had yet to be characterized.

#### 2.2.3. Packaging Module

The terminase small subunit, PBSX family phage terminase large subunit, and portal protein that made up the packaging module encoded by vB_Sags-UPM1 were 99% identical to the genes in the S73 packaging module (Figure 4a). The small terminase subunit initiated packaging, while the large terminase subunit operated as an energy-driven component that packaged and cleaved the long DNA concatemer into the empty procapsid of phage progeny via the portal protein [25]. PBSX was a defective prophage found in Bacillus subtilis that was inducible through exposure to SOS such as mitomycin-C. PBSX encoded morphogenesis genes but lacked DNA replication genes, and it has the ability to package random segments of host chromosome into virus-like particles (VLP) to be released into the environment [26].

#### 2.2.4. Morphogenesis Module

The head morphogenesis module consisting of the minor capsid, major capsid, head decoration protein, and head-tail connector protein, and the tail morphogenesis module comprising the tail completion protein, phage major tail protein (TP901-1 family), tail assembly chaperone, tail tape measure protein, tail protein, and putative glucosaminidase domain-containing tail protein, were more than 90% similar between vB_Sags-UPM1 and S73 (Figure 4a). TP901-1 was a Lactococcus lactis temperate Siphoviridae that had overall modular resemblance to *Streptococcus thermophilus* temperate Siphoviridae Sfi-21, TP-J34, and 1205 phages [22,27]. It also contained homologous DNA bases flanking the lysogeny module with other lactococcal phages, BK5-t and r1t, which allowed gene crossover between the phages [22]. The homology of the vB_Sags-UPM1 morphogenesis module with LambdaSa03, SGEHI2015-25, SG-M29 and SG-M158 was mostly observed at the putative glucosaminidase domain-containing tail protein, which might function in attachment to the host’s cell wall before degrading its peptidoglycan layer using the glucosaminidase domain to allow entry into the host’s cells (Figure 4b–f).

#### 2.2.5. Lysis Module

The lysis module consisted of endolysin and holin genes that are critical for host cell lysis. Endolysin is a peptidoglycan hydrolase enzyme that cleaves glycosidic, peptide, and/or amide bonds that hold the peptidoglycan structure. It is made up of the catalytic domain (CD) and cell wall-binding domain (CWBD) [28]. Some endolysins possess more than one CD, where each type of CD could cleave a single type of bond such as endopeptidase and amidase. The CWBD functions to bind to the components of the cell wall so that the enzymatic activity of CD could be initiated [28]. In addition, the holin protein creates an opening on the cell membrane from within, so that at the end of lytic cycle endolysin could traverse to the outer cell wall and degrade the peptidoglycan layer. The lysis module of vB_Sags-UPM1 and prophages of S73, SGEHI2015-25, SGEHI2015-95, SG-M29, and SG-M158 were 100% homologous; >90% homologous with B30, LF1, LF4, Javan7, and LambdaSa03 phages; >80% homologous with NCTC11261 (PlyGBS), Javan53, JX01, LYGO9, and LF2 phages; and <50% homologous with LF3 and A25 phages (Figure 6 and Figure 7a,b). Interestingly, the lysis module of SGEHI2015-95 was located in a region outside the prophage sequence in the host chromosome.

Holin is a highly diverse group of proteins with more than 30 distinct families that showed no conserved sequence motifs [29]. Its accumulation in the cytoplasm did not alter the cell membrane permeability throughout the lytic cycle until the “programmed” lysis period, where the holin protein oligomerized and instantaneously disrupted the cell membrane integrity [30]. This timing of lysis was subjected to intense evolutionary pressure that tinkered with the optimal lysis for different phages [29]. Figure 6a showed that vB_Sags-UPM1 holin was similar only to its closest relative, which came from prophages of S73, SGEHI2015-25, SGEHI2015-95, SG-M29, and SG-M158. Beyond that, a remarkable dissimilarity was detected even in phages that infected S. agalactiae, such as JX01 and LF2 (Figure 6a).

The endolysin of vB_Sags-UPM1, termed as Lys60, consisted of two CDs—*N*-terminal CHAP domain and a middle GH-25 domain (Figure 7c). The CWBD consisted of a C-terminal putative cell wall-binding domain SH3b (Figure 7c). The CHAP domain cleaved the d-ala-l-ala peptide bridge, while the GH-25 domain cleaved the *N*-acetyl-β-d-muramic acid glycan bond in the peptidoglycan backbone [14,31]. Experimentally, B30 and PlyGBS exhibited dual catalytic activity from both CDs, but most activity was performed by the CHAP endopeptidase when applied exogenously or “from without” [31,32]. The CHAP and GH25 domain potentially evolved from Ply187 Staphylococcal endolysin and Cpl-1 Pneumococcal endolysin, respectively [19]. The catalytic activity of Cpl-1 relied heavily on choline-binding CWBD to bind to choline-containing teichoic acid in the Pneumococcal cell wall [33]. Similarly, intact SH3b CWBD was required for the activation of B30 and the PlyGBS GH25 domain [31,32].

#### 2.2.6. Lysogeny Module

The lysogeny of vB_Sags-UPM1 was controlled by two proximally located, non-overlapping HTH domain-containing transcriptional regulator genes (Gene 51 and 52) and an integrase gene (Gene 55) downstream of Gene 52. Gene 52 was the putative cI gene due to its location upstream of the integrase gene, while Gene 51 was the putative cro gene because it was transcribed upstream of the early replication module. BLASTp could only predict Gene 55 as a functional integrase, and it only assigned the HTH domain for Genes 51 and 52, but was unable to predict excisionase in the vB_Sags-UPM1 genome. This was due to the highly diverse population of phages, even though their genome structure was markedly similar [34]. Domain prediction using Interproscan revealed that Genes 51 and 52 contained λ-like cro and cI repressors, and Gene 55 contained integrase domains derived from B. subtillis and λ phages.

### 2.3. Lytic Activity of vB_Sags-UPM1 Endolysin, Lys60

Lys60 reflected the characteristics of B30 and PlyGBS, which had been previously reported, such as the use of slightly acidic pH (pH 6.0) and CaCl_2_ addition for optimal activity [14,19]. Upon expression and purification of the recombinant Lys60, the protein showed a single band at the final elution fraction in SDS-PAGE gel (Appendix A). The enzymatic activity of Lys60 was demonstrated in the killing assay of early log phase *S. agalactiae* smyh01 and smyh02 resuspended in TRIS-NaCl buffer (pH 6.0 ± 0.1) supplemented with 10 mM of CaCl_2_ and incubated at 30 °C (Figure 5). The OD_600nm_ of Lys60-treated *S. agalactiae* smyh01 showed a reduction after 80 min of incubation in a concentration-dependent manner, compared to the untreated control (Figure 8a). Interestingly, the OD_600nm_ of Lys60-treated *S. agalactiae* smyh02 did not show a marked reduction compared to the untreated cells (Figure 9a). To confirm the antimicrobial properties of Lys60 on the host, Lys60 at different concentrations was incubated with *S. agalactiae* smyh01 and smyh02 before spreading on a BHI plate to count the percentage of viable cells compared to untreated cells. Lys60 was shown to significantly reduce the percentage of viable *S. agalactiae* smyh01 when incubated for at least 40 min (Figure 8b). However, there was no significant reduction in viable *S. agalactiae* smyh02 when incubated with Lys60 after 80 min, although a slight decrease was observed (Figure 9b).

## 3. Discussion

*S. agalactiae* outbreaks in fish farms are widespread in many countries, and have affected not only the freshwater aquaculture industry, but also increased the risk of foodborne disease transmission to humans. This study aimed to isolate a new phage targeting *S. agalactiae*, and to explore its therapeutic potential in treating *S. agalactiae* infection. Here, vB_Sags-UPM1 only infected the *S. agalactiae* tested but not the other bacteria, which was consistent with previous reports [15,18]. It possessed a morphology of *Siphoviridae* morphotype A [13], and distinct lysogenic and killing abilities in *S. agalactiae* strains, smyh01 and smyh02. *S. agalactiae* phages may present a spectrum of killing activity across different *S. agalactiae* strains, as shown in this study [4,13]. This was due to tremendous phage genome heterogeneity, which rises as a result of the mutualistic phage–host interaction that drives bacterial fitness in the environment [13]. The phages in turn, were protected inside the genome of bacterial hosts as prophages, where genetic exchange could occur more frequently and led to higher gene plasticity [35]. Consequently, this race of fitness among *S. agalactiae* strains resulted in many of the strains carrying prophage sequences in their DNA, whereby they would have the ability to not only eliminate rival strains, but also to survive in animal hosts and cause disease [35,36,37].

The genome of vB_Sags-UPM1 shared characteristics with other temperate Streptococcal *Siphoviridae* such as Sfi21, PH10, JX01, and LF1-4 in regard to the genome size, GC content, and modular organization [10,12,38,39]. The unique feature of this classification was the clustering of integrase, cro, and cI repressor genes between lysis and replication modules [22]. These phages were yet to be classified under any taxonomical genus, but their genome arrangement was closest to “λ-like” *Siphoviridae*, “L5-like” *Siphoviridae* and “P5-like” *Myoviridae* genera. The major demarcation of “Sfi21-like” genome organization from other taxa was the big gap between integrase and cro/cI repressor genes due to gene reassortment and the insertion of several foreign genes into the region [22,40]. Streptococcal lytic *Siphoviridae* such as Sfi19, Str-PAP-1, and A25 also shared the “Sfi21-like” genome structure, but they lacked the integrase gene which indicated the key role of integrase in phage lysogeny [17,38,41].

Despite the remarkable genome structure similarity of vB_Sags-UPM1 to other temperate Streptococcal *Siphoviridae*, the encoded genes were highly mosaic. This observation encapsulated the outstanding evolution of phages, where the genomic blueprint was retained while the protein expression adapted was concomitant to the evolution of bacterial hosts [42]. The genome of vB_Sags-UPM1 is closely related to the prophage of Brazilian strain S73 and prophages of Singaporean strains. S73 and SGEHI2015-95 strains were isolated from diseased fish in farm outbreaks, whereas SG-M29 was isolated in Singapore from humans who acquired the bacteria and were symptomatic [6,43,44]. These strains were found to be hypervirulent strains under serotype III, ST283 classification that had caused disease outbreaks in fishes and humans, prevalently in Southeast Asia [1]. It was recorded that the S73 strain might have originated from fish that were imported from Singapore [1]. Interestingly, the packaging and morphogenesis module of vB_Sags-UPM1 were exclusively shared by the prophage of S73, but were not homologous to the prophages of Singaporean strains, including that of fish origin, SGEHI2015-95 strain. This indicated that there was a direct evolutionary link that connected *S. agalactiae* strains of fish originating from Malaysia and Brazil. Whether or not this genetic region was transferred from Brazil to Malaysia or vice versa remains unknown. The whole genome analysis of smyh01 and smyh02 strains used in this study could provide new insights into the history of genetic remnants’ transmission across different geographical regions.

Among the modules expressed by vB_Sags-UPM1 and its related prophages, one of the most conserved regions was the lysis module. Outside of the *S. agalactiae* species, the CHAP domain shared a 27% similarity with Ply187 staphylococcal endolysin, whereas the GH25 domain was 46% similar to Cpl-1 Pneumococcal endolysin [19]. The low level of similarity in these two domains, with endolysin targeting other species, could be attributed to the wide host range of B30, PlyGBS, and PH10. For example, the PH10 phage endolysin was able to kill *S. pneumoniae* and *S. mitis* in addition to its original host, *S. oralis* [39]. Moreover, B30 and PlyGBS endolysins displayed potent lytic activity against *S. pyogenes*, group C, and group G Streptococcus. Despite that, B30 and PlyGBS displayed significantly lower activity on their original *S. agalactiae* serotypes I, II, and III hosts [14,19]. *S. agalactiae* strains that produced hyaluronate lyase, which is a virulence factor, were also resistant to B30 endolysin [14]. The reason for the low killing activity exhibited by B30 on *S. agalactiae* was the increase in peptidoglycan cross-linking and the cell wall component as the bacteria grew to a stationary phase [14]. The encapsulation of *S. agalactiae*, which differed among serotypes, might have obstructed the accessibility of the endolysins to the peptidoglycan layer [14]. It was also discovered that the lytic activity of B30 and PlyGBS, when applied exogenously, was mostly mediated by the *N*-terminal CHAP domain, whereas the GH25 domain was silent [31,32]. This suggested that the complexity of the cell wall and capsular protein of *S. agalactiae* significantly affected the efficacy of lysis by CHAP and GH25 domains in the endolysins, which could also be the case for Lys60, since it was able to efficiently kill only smyh01.

In conclusion, the isolated *S. agalactiae* phage vB_Sags-UPM1 was a temperate phage that held a close connection with the prophages of virulent strains of *S. agalactiae* serotype III, ST283 from Southeast Asia and Brazil and which was associated with fish and human diseases. The inability of vB_Sags-UPM1 to efficiently kill all *S. agalactiae* strains tested limits the application of phage therapy to treat infection in fish and humans. It was hoped that the phage lytic enzyme Lys60 would provide a new alternative to treat *S. agalactiae* infection; however, the varying susceptibility to Lys60 exhibited by different strains also restricted the application of phage endolysins to treat the disease. Nevertheless, various strategies to improve the endolysins could be carried out to increase the efficacy and host range so that its application is feasible. Moreover, the use of lytic phages described previously [18] and different types of *S. agalactiae* endolysin, such as LambdaSa01 and LambdaSa02, which contained a different wide host range CWBD called Cpl-7, could be another alternative genetic engineering strategy to treat the infection [20]. The rapid evolution of *S. agalactiae* has not only provided a challenge to come up with better treatment, but also a reminder about public health related to zoonotic diseases.

## 4. Materials and Methods

### 4.1. Bacteria Strains and Culture Condition

*Streptococcus agalactiae* smyh01 was used for phage propagation. *S. agalactiae* smyh01 and smyh02 were used for phage studies. The two *S. agalactiae* strains used in this study were isolated from diseased tilapia from ponds with disease outbreaks, and their identity was confirmed by sequencing of 16S RNA gene (Appendix A). BLAST analysis of 16S RNA gene revealed that both strains were similar to *S. agalactiae*, thus confirming their identity. The two *S. agalactiae* strains, smyh01 and smyh02, *Streptococcus pyogenes* ATCC 19615, and *Staphylococcus aureus* subsp. *aureus* PS88 (ATCC 33742) were used for the determination of host range. All bacteria were grown in brain heart infusion (BHI) media supplemented with 2 g/L yeast extract, 12 mg/L CaCl_2_, and 10 mg/L L-tryptophan in 37 °C incubation without aeration unless stated otherwise. For cloning and expression of the phage gene, plasmid pET-28a (Novagen, Houston, TX, USA) and *Escherichia coli* BL21(DE3) (Novagen, Houston, TX, USA) were used.

### 4.2. Phage Isolation

Organs (brain, spleen, and intestine) from deceased tilapia from ponds with disease outbreaks were harvested and homogenized in SM (100 mM NaCl, 8 mM MgSO_4_·7H_2_O, 50 mM Tris-Cl) buffer using mortar and pestle. The homogenized samples were filtered twice using 0.45 and 0.22 microns filters. Samples were mixed with log phase *S. agalactiae* culture (OD_600nm_ = 0.4) and incubated overnight for enrichment of phage. Bacteria and sample mixture were centrifuged at 7000× *g* and filtered using 0.45 micron filter. Single plaque was obtained by plating the phage lysates with *S. agalactiae* host using the standard double-layer agar (DLA) method. Single plaque was purified for several rounds and further enriched to increase the amount of phage lysate. Enriched phage was precipitated with polyethylene glycol-8000 (PEG8000) solution overnight at 4 °C. Phage was centrifuged at 7000× *g* and resuspended in 1/10 volume of the original phage lysate in SM buffer. Phage was filtered using 0.45 micron filter and stored at 4 °C before DNA extraction and cesium chloride (CsCl) density gradient purification.

### 4.3. One-Step Growth Curve and Kill Assay

Overnight culture of *S. agalactiae* was diluted 1:100 in fresh BHI media and grown until the OD_600nm_ reached 0.4 (10^8^ cfu/mL). At 0 min, phage stock (10^9^ pfu/mL) was added into the culture to a final MOI of 1 and incubated at 37 °C for 2 h. At 10 min interval, the culture was centrifuged, diluted 10-fold, and plated using the DLA method to measure the phage titer. In parallel, the OD_600nm_ of infected and uninfected culture was measured within a 24 h period post-infection.

### 4.4. Lysogeny Efficiency and Host Range Determination

To determine the host range of *S. agalactiae* phage, lawns of bacteria (*S. agalactiae* smyh01, *S. agalactiae* smyh02, *S. pyogenes*, and *S. aureus*) were prepared following the DLA method. A total of 10 µL of 10^9^ pfu/mL phage stock was spotted on the bacterial lawns and incubated overnight to observe the presence of clearing zones. The lysogeny efficiency of *S. agalactiae* phage in both hosts was performed following the methods as described previously [45]. Briefly, 100 µL of 10^9^ pfu/mL phage stock was spread uniformly on BHI agar, dried, and denoted as phage-seeded plates. A log phase of *S. agalactiae* culture (OD_600nm_ = 0.7) was serially diluted 10-fold and spread uniformly on the phage plates and left to dry. Similarly, the bacterial dilution was also spread on BHI agar only and denoted as control plates. The lysogeny efficiency was calculated by dividing the CFU on phage plate with CFU on control plates from the same countable dilution and multiplied by 100.
(1)$$\frac{CFU\;of\;phage\;seeded\;plate}{CFU\;of\;control\;plate}\times 100=Lysogeny\;efficiency\;(\%)$$

### 4.5. Transmission Electron Microscopy (TEM)

Phages were purified using CsCl density gradient centrifugation by layering CsCl solution at density of 1.7, 1.5, 1.4, and 1.3 g/mL, followed by 2 mL of precipitated phage. The sample was centrifuged at 250,000× *g* for 2 h at 4 °C using Class S ultracentrifuge in SW40 Ti Swinging-Bucket Rotor (Beckman Coulter Inc., Brea, CA, USA). The bluish-white band containing phage particles was aspirated and dialyzed twice using 1000× volume of SM buffer. The purified sample was adsorbed onto Carbon Coated Fomvar Grid and stained with 1% Uranyl Acetate. The phage morphology was viewed using JEM21000F TEM, 200 kV Field Emission (JEOL, Tokyo, Japan).

### 4.6. Phage DNA Extraction

Prior to phage genomic DNA extraction, PEG-precipitated phage was treated with 0.8 U/mL final concentration of DNase I (Thermo Scientific, Waltham, MA, USA) and 0.1 mg/mL RNase A (EMD Millipore Corp., Single Oak, CA, USA) for 30 min to remove host’s DNA and RNA. Phage capsule and proteins were digested using 50 µg/mL final concentration of Proteinase K (iNtRON Biotechnology, Seongnam, Republic of Korea) and 1% sodium dodecyl-sulphate for 60 min at 55 °C. Equal volume of phenol-chloroform-isoamyl alcohol (25:24:1) was mixed with the sample and vortexed briefly. The mixture was centrifuged at 16,000× *g* for 5 min and the aqueous phase was transferred to a new tube. DNA was precipitated using equal volume of 100% ethanol and centrifuged at 16,000× *g* for 10 min. The pellet was washed with 70% ethanol twice. The DNA was air-dried and resuspended in TE buffer for DNA quantification and whole genome sequencing.

### 4.7. Bioinformatics

#### 4.7.1. Illumina Library Preparation and Genome Sequencing

Approximately 100 ng of DNA as measured by Denovix High Sensitivity Kit (Denovix, Wilmington, DE, USA) was fragmented to 350 bp using a Bioruptor followed by NEB Ultra II library preparation according to the manufacturer’s instructions (NEB, Ipswich, MA, USA). Sequencing was performed on a NovaSEQ6000 (Illumina, San Diego, CA, USA) generating approximately 1 Gb of paired-end data (2 × 150 bp) for each sample.

#### 4.7.2. De Novo Assembly—Illumina

Raw Illumina paired-end reads were trimmed with fastp v0.21 [46] to remove low-quality bases and Illumina adapter sequences. A small phage genome (<100 kb) that has been enriched (DNAse-treated, 10^9^ pfu/mL PEG-precipitated phage) will have an extremely high genomic coverage if the original amounts of datasets were to be used, leading to unnecessary complication of the assembly graph. Therefore, prior to de novo assembly, the trimmed reads were subsampled randomly using the seqtk “sample” function to sample only 10% of the read fraction. The subsampled paired-end reads were used for de novo assembly in SPAdes v3.15.0 (--isolate setting) [47]. From the de novo assembled contigs, phage-derived contigs were identified using PHASTER (https://phaster.ca/) (accessed on 5 October 2021) (Appendix A) [48]. Extraction of the contigs showing 1 × 10^−100^ E-values to the PHASTER database were subsequently extracted with the seqtk subseq function (https://github.com/lh3/seqtk) (accessed on 5 October 2021) and analyzed further.

#### 4.7.3. Genome Annotation

Protein-coding gene prediction was performed using Glimmer3 v1.5 [49] and Prodigal v2.6 [50], generating two files corresponding to the predicted genes and the corresponding putative proteins.

#### 4.7.4. Bioinformatics Analysis

The whole phage genome was searched for the closest relative using BLASTn against the default nucleotide collection (nr/nt) and virus database (taxid: 10239). Average nucleotide identity (ANI) of a phage was calculated using JSpecies with a cutoff of 95% to indicate similar taxonomic affiliation [51]. The putative proteins were uploaded to open-source webserver eggnog mapper (http://eggnog-mapper.embl.de/) (accessed on 5 October 2021) [52] and BLASTp for functional prediction. The protein domains were predicted using InterPro open-web database [53]. Phylogenetic tree of whole vB_Sags-UPM1 genome was constructed using Neighbor Joining method [54], which was available in NCBI and visualized using Interactive Tree of Life (iTOL) web-server [55]. Geneious Prime (Version 2023.0.4) was used for ProgressiveMauve [56] and ClustalOmega [57] alignment to compare vB_Sags-UPM1 genome with other phages, and to compare nucleic acid sequence of holin and endolysin genes of vB_Sags-UPM1 with similar genes of other phages. The holin and endolysin nucleic acid phylogenetic trees were analyzed using PhyML (Version 3.3), which was based on maximum likelihood (ML) method [58] using default parameters in Geneious Prime software. The whole genome map of vB_Sags-UPM1 and all pairwise comparisons were visualized using Geneious Prime.

### 4.8. Cloning and Expression of Endolysin Gene

A pair of forward primer containing NcoI site (5′-GCCCCATGGCAACTTATCAAGAATA-3′) and reverse primer containing XhoI site (5′-CCGCTCGAGCGGCATATCTGTTGCATCA-3′) was used to amplify the Lys60 gene following the PCR reaction—4 µL 5× Phusion HF Buffer (Thermo Scientific, Waltham, MA, USA), 1 µL 10 µM forward primer, 1 µL 10 µM reverse primer, 0.4 µL 10 mM dNTP mix, 0.2 µL Phusion™ Plus DNA Polymerase (Thermo Scientific, Waltham, MA, USA), and 1 µL 100 ng/µL DNA template and 12.4 µL H_2_O. The PCR was performed using Eppendorf^®^ Mastercycler^®^ Nexus Thermal Cycler (Eppendorf, Hamburg, Germany) set to 98 °C for 2 min (initial denaturation); 35 cycles of 98 °C for 1 min, 57 °C for 30 s, and 72 °C for 1 min (amplification); and 72 °C for 5 min (final extension). PCR amplicons and plasmid pET were digested with NcoI and XhoI restriction enzymes (Thermo Scientific, Waltham, MA, USA) for 1 h at 37 °C before ligation using T4 Ligase (Thermo Scientific, Waltham, MA, USA) at 22 °C for 1 h. Recombinant DNA was introduced into *E. coli* using the traditional heat shock method. Positive clones were selected on LB with kanamycin plates and screened using colony PCR using forward (5′-TCCCGCGAAATTAATACGAC-3′) and reverse primers (5′-TATGCTAGTTATTGCTCAGC-3′) that recognized regions of pET-28a that flanked the multiple cloning site. For gene induction, overnight culture of recombinant *E. coli* was added into fresh LB medium and incubated with shaking at 37 °C. Upon reaching an OD_600nm_ of 0.4, a final concentration of 0.5 mM IPTG was added into the culture and incubated at 18 °C for 24 h. Cells were harvested and washed using buffer (50 mM TRIS, 150 mM NaCl, pH 6.0) and sonicated for eight minutes with a resting period of 30 s. Soluble protein in the supernatant was applied into cOmplete His-Tag Purification Resin (Merck, Darmstadt, Germany) to purify proteins that contained His tag. The eluted protein was concentrated in buffer using Pierce™ Protein Concentrator with a polyethersulfone membrane, 10 K molecular weight cutoff (Thermo Scientific, Waltham, MA, USA). The protein concentration was quantified using Pierce^®^ BCA Protein Assay Kit (Thermo Scientific, Waltham, MA, USA) following the manufacturer’s protocol. The presence of Lys60 in crude as well as in purified fractions was detected using 12% SDS-PAGE with 10–180 kDa PageRuler™ Prestained Protein Ladder (Thermo Scientific, Waltham, MA, USA), followed by staining with Coomassie blue or Western Blot. After protein transfer from gel to Amersham Hybond polyvinylidene difluoride membrane (GE Healthcare Life Science, Piscataway, NJ, USA), the membrane was stained with 1:1000 6× His-tag monoclonal antibody (Thermo Scientific, Waltham, MA, USA) followed by staining with 1:10,000 goat anti-mouse IgG secondary antibody, alkaline phosphatase (AP) (Thermo Scientific, Waltham, MA, USA). The substrate 5-Bromo-4chloro-3-indolyl-phospate (BCIP) was used, which reacted with nitro blue tetrazolium (NBT) for detection. The gel and Western Blot membrane was viewed using Amersham Imager 600 (GE Healthcare, Chicago, IL, USA).

### 4.9. Lys60 Lytic Activity

The lytic activity of Lys60 was determined using turbidity reduction assay of *S. agalactiae* at 600 nm absorbance. Briefly, the concentration of proteins was standardized to 645.6 µg/mL. Serial dilutions of 484.2 µg/mL, 322.8 µg/mL, 242.1 µg/mL, 161.4 µg/mL, and 80.7 µg/mL were prepared in buffer (50 mM TRIS, 150 mM NaCl, pH 6.0). Fifty mL of the early log phase of *S. agalactiae* culture (OD_600nm_ = 0.2) was centrifuged, washed, and resuspended in 5 mL final volume of buffer. A volume of 178 µL of bacteria was mixed with 2 µL, 1M CaCl_2_ (final concentration of 10 mM) and dispensed into the 96-well plate in triplicate for each concentration of Lys60. A total of 20 µL of protein dilutions and empty buffer was added into bacterial suspensions, mixed, and incubated at 30 °C. At 5 min intervals, the absorbance at OD_600nm_ was measured for a period of 2 h using iMark™ Microplate Reader (Bio-Rad, Hercules, CA, USA). Additionally, the antimicrobial activity was determined by calculating the percentage of viable bacteria relative to control after treating with Lys60 (64.56, 32.28, and 8.07 µg/mL final concentration) or untreated for 0, 40, and 80 min. The CFU/mL of treated and untreated bacteria was determined and the percentage viability at each time point post-treatment was calculated based on the following formula.
(2)$$\frac{CFU/mL\;of\;Lys60\;treated\;bacteria}{CFU\;of\;untreated\;bacteria}\times 100=\%CFU/mL\;of\;control$$

### 4.10. Data Analysis

The data were analyzed using GraphPad Prism 8. Firstly, the normality of datasets was determined using Shapiro–Wilk test, whereby most data were normally distributed (parametric data). For the comparison of burst size, absorbance at OD_600nm_, and lysogeny efficiency between *S. agalactiae* smyh01 and smyh02, the data were analyzed using unpaired *t*-test where a *p* value of <0.05 was considered statistically significant. For the comparison of antimicrobial activity of Lys60 endolysin on *S. agalactiae* smyh01 and smyh02 between different time points post-infection, and between different enzyme concentration, the data were analyzed using two-way ANOVA with Tukey’s post hoc test, where a *p* value of <0.05 was considered statistically significant.

## Figures and Tables

**Figure 1 pharmaceuticals-16-00698-f001:**
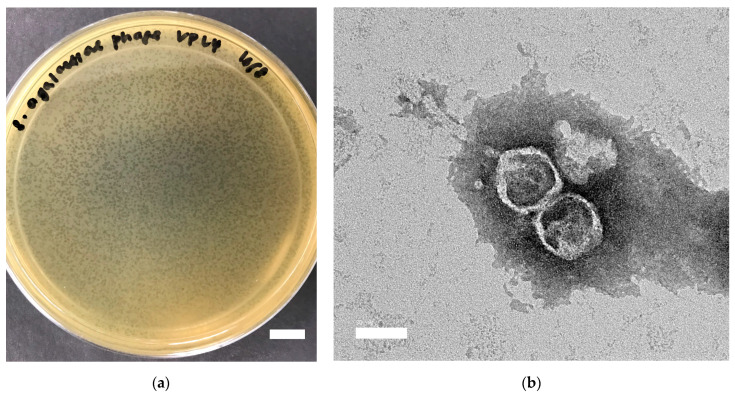
(**a**) *S. agalactiae* phage vB_Sags-UPM1 plaques on lawn of smyh01 and (**b**) morphology under transmission electron microscopy (TEM). Scale bar indicates 10 mm and 50 nm for (**a**,**b**), respectively.

**Figure 2 pharmaceuticals-16-00698-f002:**
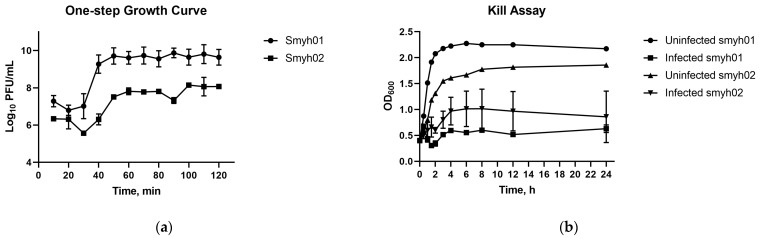
(**a**) One-step growth curve and (**b**) kill assay of vB_Sags-UPM1 in *S. agalactiae* smyh01 and smyh02. The data are represented as mean ± standard deviation from three independent replicates.

**Figure 3 pharmaceuticals-16-00698-f003:**
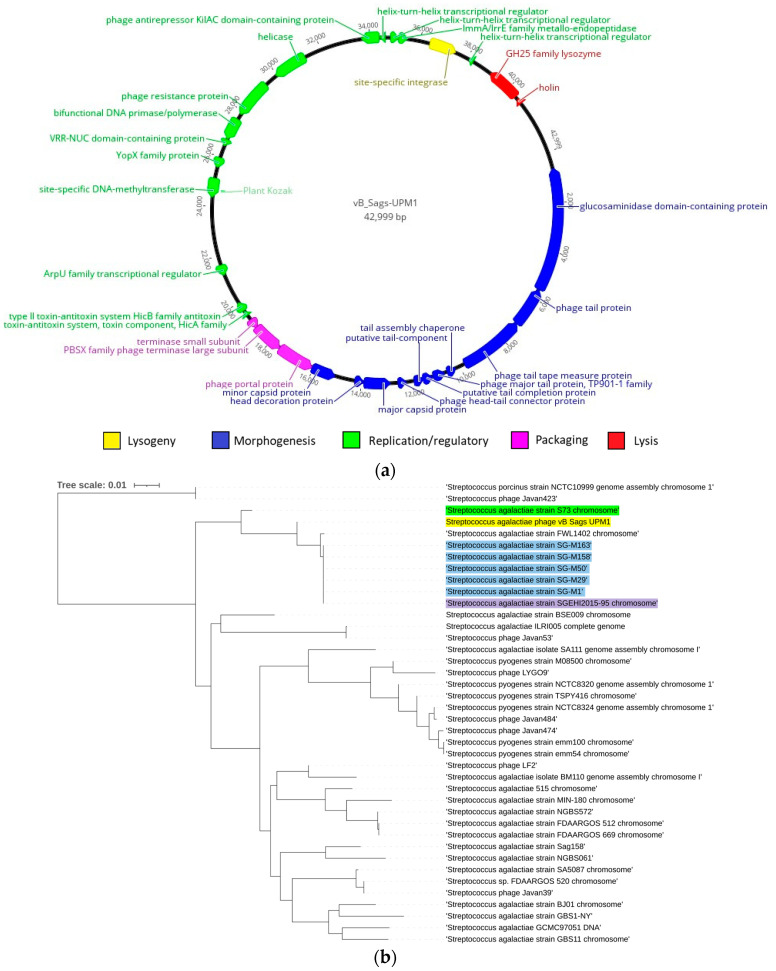
(**a**) The whole genome map of *S. agalactiae* phage vB_Sags-UPM1. (**b**) Phylogenetic tree of vB_Sags-UPM1 with *S. agalactiae* and *S. pyogenes* prophages and temperate phages based on Neighbor Joining method. Yellow-highlighted leaf indicates vB_Sags-UPM genome; green-highlighted leaf indicates *S. agalactiae* Brazilian fish isolate S73 genome; blue-highlighted leaf indicates *S. agalactiae* Singaporean human isolates genomes; and purple-highlighted leaf indicates *S. agalactiae* Singaporean fish isolate SGEHI2015-95 genome.

**Figure 4 pharmaceuticals-16-00698-f004:**
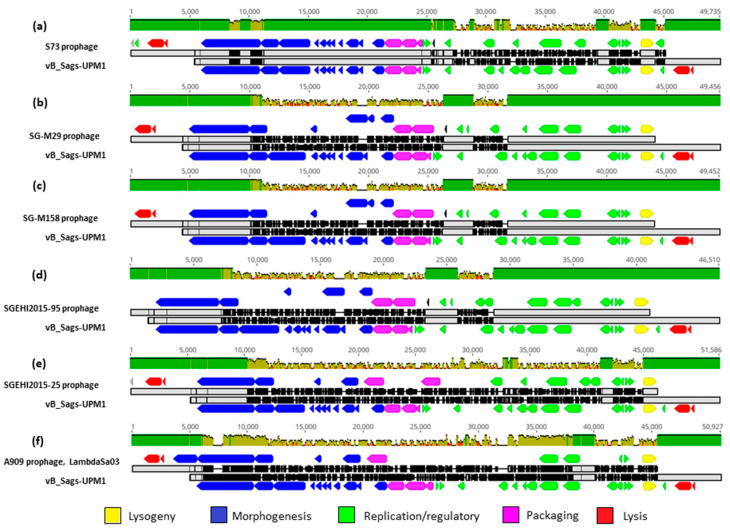
ClustalOmega nucleic acid pairwise alignment of vB_Sags-UPM1 with prophages of S73 (**a**), SG-M29 (**b**), SG-M158 (**c**), SGEHI2015-95 (**d**), SGEHI2015-25 (**e**), and A909 (LambdaSa03) (**f**). The gray- and black-shaded area represent regions of homology and dissimilarity, respectively, in the nucleic acid sequences. The green, brown, and red bars on top of alignment indicate >99%, 30–70%, and <30% nucleic acid similarity, respectively.

**Figure 5 pharmaceuticals-16-00698-f005:**
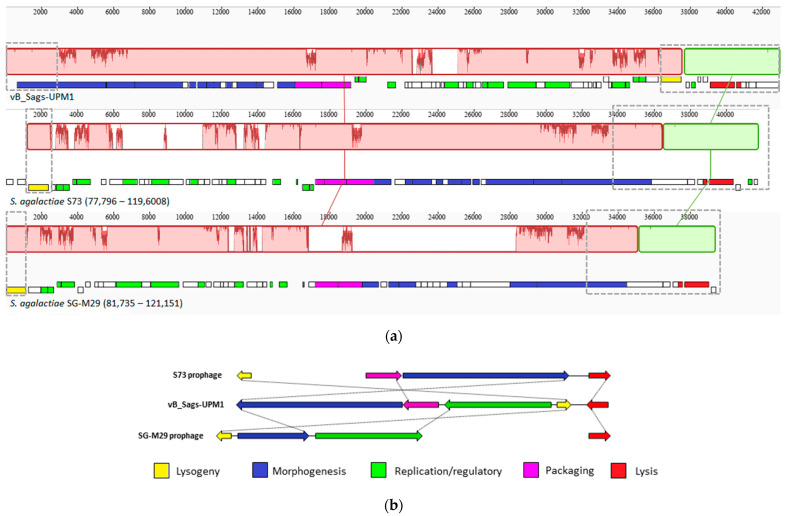
(**a**) Whole genome progressiveMauve alignment of vB_Sags-UPM1 and prophages of S73 and SG-M29 isolates. The bar graph of matching colors with connecting lines indicate the degree of homology between different genomes. The coding sequences of each genome are depicted below the bar graph with different color-shaded rectangles to indicate different phage modules, and a white rectangle indicates genes of unknown function. Gray dashed-boxes indicate the conserved region shared among closely related prophages with vB_Sags-UPM1. (**b**) The summary of vB_Sags-UPM1 whole genome relatedness with prophages S73 and SG-M29. The dotted lines connecting different modules indicate homology between sequences. The lysis and lysogeny modules are shared by all three sequences; the packaging and morphogenesis module of the S73 prophage are homologous to vB_Sags-UPM1; the replication/regulatory module and part of the tail morphogenesis module of SG-M29 are homologous to vB_Sags-UPM1.

**Figure 6 pharmaceuticals-16-00698-f006:**
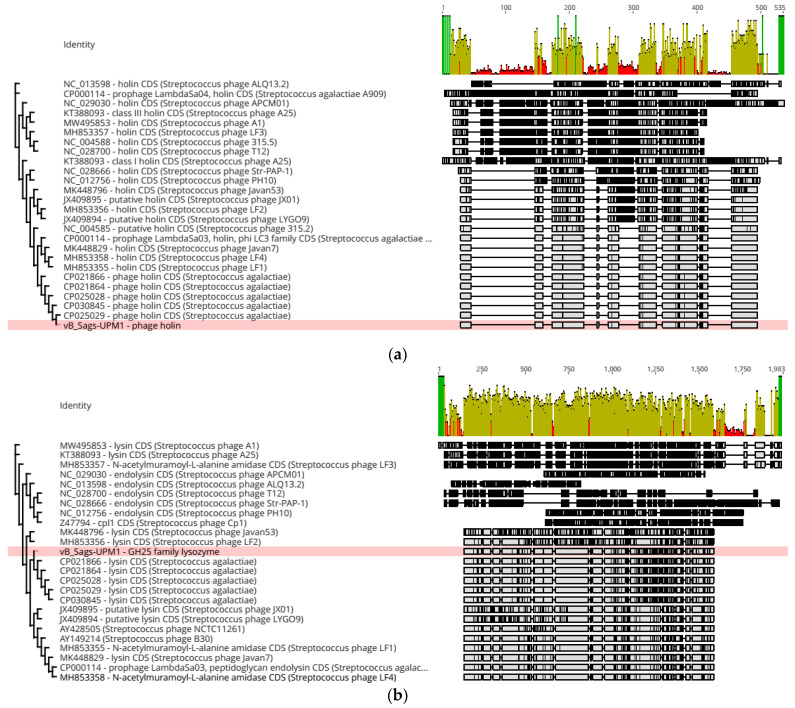
Phylogenetic tree of holin (**a**) and endolysin (**b**) genes of vB_Sags-UPM1 and other related prophages. Tree was constructed using PhyML plugin in Geneious Prime which was based on the maximum likelihood (ML) method. Multiple sequence alignment was depicted on the right side of the figure, where the gray- and black-shaded area represent regions of homology and dissimilarity, respectively, in the nucleic acid sequences. The green, brown, and red bars on top of alignment indicate >99%, 30–70%, and <30% nucleic acid similarity, respectively. Pink highlight indicates vB_Sags-UPM1 holin and endolysin (Lys60).

**Figure 7 pharmaceuticals-16-00698-f007:**
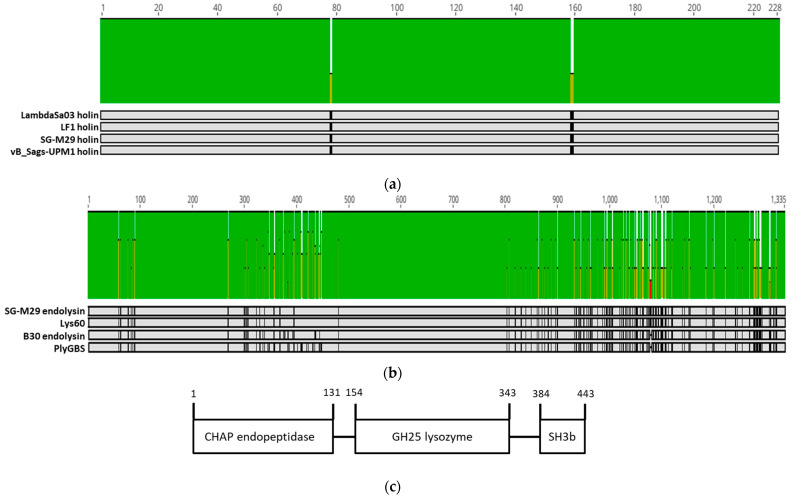
ClustalOmega multiple sequence alignment of holin (**a**) and endolysin (**b**) genes from vB_Sags-UPM1 and other closely related prophages and temperate phages. The gray- and black-shaded area represent regions of homology and dissimilarity, respectively, in the nucleic acid sequences. The green, brown, and red bars on top of alignment indicate >99%, 30–70%, and <30% nucleic acid similarity, respectively. (**c**) The predicted protein domains of SG-M29 prophage, Lys60, B30, and PlyGBS endolysins using Interproscan web-server. The numbering above domains indicates positions of amino acids. CHAP: cysteine, histidine-dependent amidohydrolases/peptidases; GH25: glycoside hydrolase family 25; SH3b: SRC Homology 3b.

**Figure 8 pharmaceuticals-16-00698-f008:**
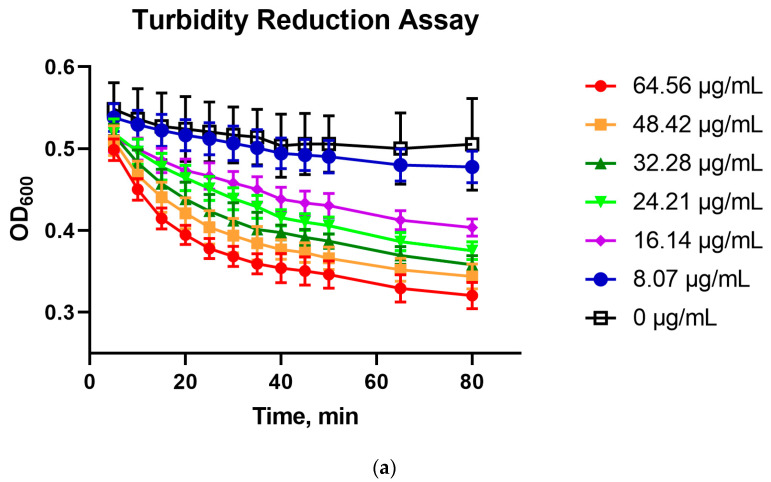
Turbidity reduction (**a**) and antimicrobial assay (**b**) of Lys60 at different concentrations on *S. agalactiae* smyh01. The data are represented as mean ± standard deviation from three independent replicates, where ‘*’ and ‘**’ indicate *p* values of <0.05 and <0.01, respectively (Two-way ANOVA with Tukey’s post hoc test).

**Figure 9 pharmaceuticals-16-00698-f009:**
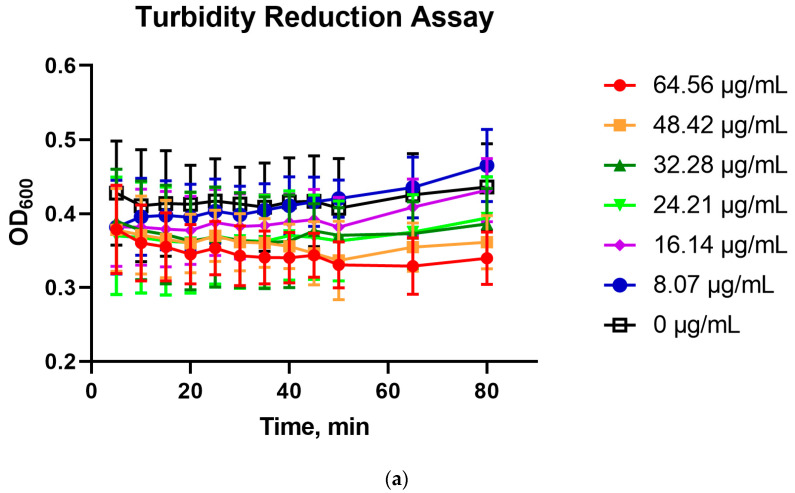
Turbidity reduction (**a**) and antimicrobial assay (**b**) of Lys60 at different concentrations on *S. agalactiae* smyh02. The data are represented as mean ± standard deviation from three independent replicates where no significant difference was detected (Two-way ANOVA with Tukey’s post hoc test).

**Table 1 pharmaceuticals-16-00698-t001:** Genes encoded by vB_Sags-UPM1.

Gene	Start–End	Strand	Accession No.	BLASTp Description [Organism]	AA Identity	Accession AA Length	Family/Domain (InterPro ID)	Function
1	1–651	−	WP_224208974	DUF859 family phage minor structural protein [*S. agalactiae*]	215/217 (99%)	504	Putative viral structural protein, DUF859 (IPR008577)	−
2 ^a^	652–5619	−	WP_053515050	Glucosaminidase domain-containing protein [*S. agalactiae*]	1653/1655 (99%)	1655	Mannosyl-glycoprotein endo-beta-*N*-acetylglucosaminidase (IPR002901), CHAP (IPR007921)	Tail protein
3 ^b^	5610–7172	−	PPQ23787	Phage tail protein [*S. agalactiae*]	519/520 (99%)	520	Siphovirus-type tail component (IPR008841)	Tail protein
4	7173–9821	−	WP_017647846	Phage tail tape measure protein [*S. agalactiae*]	882/882 (100%)	882	Phage tail tape measure protein (IPR010090), Rhodanese-like domain (IPR001763)	Tail protein
5 ^a^	9814–10,119	−	EPU00268	Hypothetical protein SAG0108_10345 [*S. agalactiae* BSU92]	101/101 (100%)	103	−	−
6	10,194–10,553	−	MCK6349393	Tail assembly chaperone [*S. agalactiae*]	118/119 (99%)	121	Phage tail assembly chaperone (TAC) proteins (IPR024410)	Tail protein
7	10,669–11,154	−	WP_017647848	Phage major tail protein, TP901-1 family [*S. agalactiae*]	161/161 (100%)	161	Phage major tail protein TP901-1 (IPR011855)	Major Tail
8	11,163–11,549	−	WP_017647849	DUF3168 domain-containing protein [*S. agalactiae*]	128/128 (100%)	128	Tail completion protein (IPR021508)	Tail protein
9	11,546–11,929	−	WP_000609260	HK97 gp10 family phage protein [*S. agalactiae*]	104/126 (83%)	126	Bacteriophage HK97-gp10, putative tail-component (IPR010064)	Tail protein
10	11,922–12,221	−	WP_017647851	Hypothetical protein [*S. agalactiae*]	99/99 (100%)	99	−	−
11	12,218–12,571	−	WP_017647852	Phage head-tail connector protein [*S. agalactiae*]	117/117 (100%)	117	Phage gp6-like head-tail connector protein (IPR021146)	Head-tail connector
12 ^a^	12,587–12,841	−	WP_001229661	HeH/LEM domain-containing protein [Streptococcus]	83/84 (99%)	84	HeH/LEM domain (IPR025856), Rho termination factor (IPR036269), SAP domain (IPR036361)	DNA/RNA-binding protein
13	12,854–13,906	−	WP_001185213	Major capsid protein [*S. agalactiae*]	350/350 (100%)	350	Major capsid protein GpE (IPR005564)	Major capsid
14	13906–14289	−	WP_001042771	Head decoration protein [Streptococcus]	127/127 (100%)	127	Head decoration protein D (IPR004195)	Head protein
15	14,299–14,877	−	WP_017647853	DUF4355 domain-containing protein [*S. agalactiae*]	192/192 (100%)	192	Protein of unknown function DUF4355 (IPR025580)	−
16	15,094–16,065	−	WP_017647854	Minor capsid protein [*S. agalactiae*]	323/323 (100%)	323	Phage head morphogenesis domain (IPR006528)	Head protein
17	16,049–17,542	−	WP_017647855	Phage portal protein [*S. agalactiae*]	497/497 (100%)	497	Portal protein (IPR021145)	Portal protein
18	17,555–18,796	−	WP_017647856	PBSX family phage terminase large subunit [*S. agalactiae*]	413/413 (100%)	413	Bacteriophage terminase, large subunit (IPR006437)	Terminase
19 ^b^	18,783–19,154	−	MCC9692304	Terminase small subunit [*S. agalactiae*]	123/123 (100%)	150	Terminase small subunit (IPR005335)	Terminase
20	19,388–19,573	+	KXA53940	Toxin-antitoxin system, toxin component, HicA family [*S. agalactiae*]	61/61 (100%)	84	HicA mRNA interferase family (IPR012933)	RNA-binding protein
21	19,625–20,002	+	WP_017647857	Type II toxin-antitoxin system HicB family antitoxin [*S. agalactiae*]	125/125 (100%)	125	HicB-like antitoxin of toxin-antitoxin system (IPR031807)	RNA-binding protein
22	21,203–21,637	−	WP_000142570	ArpU family transcriptional regulator [Streptococcus]	144/144 (100%)	144	Putative autolysin regulatory protein ArpU-like (IPR006524)	Transcriptional regulator
23	22,023–22,163	−	AFQ95908	Hypothetical protein [Streptococcus phage LYGO9]	46/46 (100%)	46	−	−
24	22,163–22,330	−	AFQ95925	Hypothetical protein [Streptococcus phage LYGO9]	55/55 (100%)	55	−	−
25	22,327–22,545	−	AFQ95980	Hypothetical protein [Streptococcus phage JX01]	72/72 (100%)	81	−	−
26	22,542–23,078	−	PPQ23766	Hypothetical protein C4888_10100 [*S. agalactiae*]	178/178 (100%)	179	HNH nuclease (IPR003615), NUMOD4 (IPR010902)	DNA-binding protein
27	23,083–23,643	−	PPQ23765	Hypothetical protein C4888_10095 [*S. agalactiae*]	185/186 (99%)	186	Protein of unknown function DUF1642 (IPR012865)	−
28	23,633–23,875	−	PPQ23764	Hypothetical protein C4888_10090 [*S. agalactiae*]	80/80 (100%)	80	−	−
29	23,865–24,104	−	PPQ23763	Hypothetical protein C4888_10085 [*S. agalactiae*]	79/79 (100%)	79	−	−
30	24,171–24,332	−	WP_196755760	Hypothetical protein [*S. parauberis*]	51/53 (96%)	57	−	−
31	24,380–25,129	−	PPQ23762	Site-specific DNA-methyltransferase [*S. agalactiae*]	249/249 (100%)	249	Restriction/modification DNA-methyltransferase (IPR001091)	DNA methylase
32	25,132–25,401	−	WP_053515031	Hypothetical protein [*S. agalactiae*]	89/89 (100%)	89	−	−
33	25,398–25,562	−	WP_000159367	Hypothetical protein [*S. agalactiae*]	54/54 (100%)	54	−	−
34	25,559–25,939	−	WP_053515030	YopX family protein [*S. agalactiae*]	126/126 (100%)	126	YopX protein (IPR019096)	Bacterial type III secretion system
35	25,936–26,094	−	WP_017646174	Hypothetical protein [Streptococcus]	52/52 (100%)	52	−	−
36	26,078–26,374	−	WP_000763916	Nucleotide modification associated domain-containing protein [*S. agalactiae*]	98/98 (100%)	98	Nucleotide modification associated domain 1 (IPR011630)	−
37 ^b^	26,453–26,803	−	ASA80945	VRR-NUC domain-containing protein [*S. agalactiae*]	115/116 (99%)	116	VRR-NUC domain (IPR014883)	Nuclease
38	26,763–27,629	−	WP_053515029	Bifunctional DNA primase/polymerase [*S. agalactiae*]	288/288 (100%)	288	DNA primase/polymerase, bifunctional, *N*-terminal (IPR015330), Primase, C-terminal 1 (IPR014820)	DNA polymerase/primase
39	27,874–29,427	−	ALB15147	Phage resistance protein [*S. agalactiae*]	517/517 (100%)	517	−	−
40	29,445–29,927	−	WP_053515027	DUF669 domain-containing protein [*S. agalactiae*]	160/160 (100%)	160	Protein of unknown function DUF669 (IPR007731)	−
41	29,940–31,307	−	ASA80941	Helicase [*S. agalactiae*]	455/455 (100%)	468	Helicase, C-terminal (IPR001650), Helicase/UvrB, *N*-terminal (IPR006935), Helicase superfamily 1/2, ATP-binding domain (IPR014001)	Helicase
42	31,382–32,065	−	WP_000704951	AAA family ATPase [*S. agalactiae*]	227/227 (100%)	227	Phage nucleotide-binding protein (IPR006505)	DNA/RNA-binding protein
43	32,052–32,342	−	WP_000650504	Hypothetical protein [Streptococcus]	96/96 (100%)	96	−	−
44	32,339–32,539	−	WP_001058282	Hypothetical protein [Streptococcus]	66/66 (100%)	66	−	−
45	32,532–32,621	−	AYJ74898	Hypothetical protein [Streptococcus phage LF1]	29/29 (100%)	29	−	−
46	32,615–32,758	−	WP_172798516	Hypothetical protein [*S. agalactiae*]	47/47 (100%)	47	−	−
47	32,788–33,045	−	WP_001191791	Hypothetical protein [Streptococcus]	85/85 (100%)	85	−	−
48	33,174–33,467	+	WP_001104373	Hypothetical protein [*S. agalactiae*]	97/97 (100%)	97	−	−
49	33,464–33,622	−	WP_017647876	Hypothetical protein [*S. agalactiae*]	52/52 (100%)	52	−	−
50	33,655–34,380	−	WP_053515025	Phage anti-repressor KilAC domain-containing protein [*S. agalactiae*]	241/241 (100%)	241	AntA/AntB anti-repressor (IPR013557), Anti-repressor protein, C-terminal (IPR005039)	−
51	34,409–34,621	−	WP_053515024	Helix-turn-helix transcriptional regulator [*S. agalactiae*]	70/70 (100%)	70	Cro/C1-type helix-turn-helix domain (IPR001387)	Cro/CI repressor
52	34,817–35,158	+	WP_053515023	Helix-turn-helix transcriptional regulator [*S. agalactiae*]	113/113 (100%)	113	Cro/C1-type helix-turn-helix domain (IPR001387)	Cro/CI repressor
53	35,151–35,519	+	WP_053515022	ImmA/IrrE family metallo-endopeptidase [*S. agalactiae*]	122/122 (100%)	125	IrrE *N*-terminal-like domain (IPR010359)	Metallopeptidase
54	35,528–36,214	+	WP_053515021	Hypothetical protein [*S. agalactiae*]	228/228 (100%)	228	−	−
55	36,388–37,485	+	MBY5048373	Site-specific integrase [*S. agalactiae*]	365/365 (100%)	384	AP2-like integrase, *N*-terminal domain (IPR028259), Integrase, catalytic domain (IPR002104), Integrase, SAM-like, *N*-terminal (IPR004107)	Integrase
56	37,755–37,937	−	WP_017647839	Hypothetical protein [*S. agalactiae*]	60/60 (100%)	60	−	−
57	38,060–38,269	−	WP_000424774	Helix-turn-helix transcriptional regulator [*S. agalactiae*]	69/69 (100%)	69	Cro/C1-type helix-turn-helix domain (IPR001387)	Cro/CI repressor
58	38,419–38,565	+	WP_001030869	Hypothetical protein [*S. agalactiae*]	48/48 (100%)	48	−	−
59	38,706–38,957	+	WP_000455662	Hypothetical protein [*S. agalactiae*]	83/83 (100%)	83	−	−
60	39,101–40,435	−	WP_017647840	GH25 family lysozyme [*S. agalactiae*]	444/444 (100%)	444	CHAP domain (IPR007921), Glycoside hydrolase, family 25 (IPR002053), putative SH3 domains	Endolysin
61	40,561–40,788	−	WP_000609111	Phage holin [Streptococcus]	75/75 (100%)	75	Bacteriophage holin (IPR006485)	Holin
62	40,781–41,083	−	WP_017647841	Hypothetical protein [*S. agalactiae*]	100/100 (100%)	100	−	−
63	41,092–41,292	−	AMQ13761	Hypothetical protein CUGBS08_00165 [*S. agalactiae*]	66/66 (100%)	66	−	−
64	41,243–41,635	−	WP_000889196	DUF1366 domain-containing protein [Streptococcus]	130/130 (100%)	130	Protein of unknown function DUF1366 (IPR009796)	−
65 ^c^	41,647–42,999	−	WP_053515051	DUF859 family phage minor structural protein [*S. agalactiae*]	450/450 (100%)	667	Putative viral structural protein DUF859 (IPR008577)	−

Legend: ^a^ indicates codons started with GTG; ^b^ indicates codons started with TTG; ^c^ indicates codons started with CCG.

## Data Availability

The whole genome sequence applicable in this article can be found in NCBI under accession number: OQ633473.

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
