# Peer review of "Comprehensive Characterization of a Streptococcus agalactiae Phage Isolated from a Tilapia Farm in Selangor, Malaysia, and Its Potential for Phage Therapy"

_pharmaceuticals, 2023, doi:10.3390/ph16050698_

Round 1

Reviewer 1 Report

The authors report the isolation of a Streptococcus agalactiae phage from infected tilapia, named vB_Sags-UPM1. They have performed whole genome sequencing and discovered that it shares homology with S. agalactiae S73 chromosome and other S. agalactiae strains. The presence of integrase gene confirms that this is a temperate phage. Importantly, the phage is able to lyse 2 isolated strains of S. agalactiae isolated from affected fish, while sparing other bacteria. The authors have also recombinantly expressed and purified the protein product of phage endolysin gene, named Lys60. They show lytic activity for one of the isolated S. agalactiae strains, smyh1, as a decrease in culture turbidity and as a result of an antimicrobial assay.

They present a very thorough bioinformatic study that describes the homology of different structural, regulatory, and virulence elements of the newly discovered phage to other phages. Although the genome structure is similar to other temperate streptococcal Siphoviridae phages, the genomic elements show a mosaic organization and authors propose that this can be a result of recombination events between prophage of Brazilian strain S73 with prophages of Singapore strains SG-M29 and SGEHI2015-95.

The research into the field of potentially lytic phages to be used as a treatment in S. agalactiae infection is extremely important in today’s quest for novel treatment options. Although the novel isolate is a temperate phage and the isolated Lys60 was not affecting both tested Streptococcus isolates, this report contributes to the knowledge base on potential novel therapeutic methods, even if the elements presented here should be further developed for this purpose.

The article is clearly written and the inference is well presented. Graphical data presentation could be improved in some points (see specific comments below). Please find below a list of remarks, which I hope you find helpful in revision.  

Page 1 line 44: „The outbreak…“ – this sentence is not so clear. Maybe: “The outbreak of S. agalactiae in fish farms are widespread in … and caused predominantly by serotype III…implicated in fish-borne infection…“ or similar.

Figure 2b: Error bars on the curves for uninfected cultures?

Figure 3b is hardly legible and the font should be enlarged.

Line 175: “conservation was observed at least in the lysogeny, lysis, and the end region…”: I would propose to reword this sentence as the authors want to express that these regions were most conserved.

Line 192: with prophages S73 and SG-M29

Line 211: Listeria monocytogenes in italics

Line 278: CDs (throughout the text)

Line 284: “applied “from without””: could you please explain what “from without” is?

Line 317: killing assay

Figure 9a: the symbols are difficult to tell from each other: maybe introduce color code?

Figure 8b and 9b: there is not much difference between both shades of grey

Line 339: outbreaks in fish farms

Line 431: Supplementary file 3: The raw sequencing data are provided, instead (or in addition) a concluding statement should be included in the main text at this point.

Line 450: at 4°C

Line 564: which substrate was used for detection of the western blot?

Supplementary Figure 2: uncut images should be shown.

Reviewer 2 Report

The current study ‘Comprehensive characterization of a Streptococcus agalactiae phage isolated from a tilapia farm in Selangor, Malaysia and its potential for phage therapy’ helps to explore a new alternatives in treating S. agalactiae infection in fish by using phages or phage-encoded endolysins, using bioinformatics tools to compare its origin and to isolate the putative endolysin gene. The research work is fundamentally sound and targets to develop of new antimicrobial alternatives to treat S. agalactiae infection, specifically in tilapia. However, some minor issues must be addressed by emphasizing the following points before this work is accepted.

1.      The correct result description was provided, making the findings more effective, but the results were presented in somewhat mixed detail. When the results are combined, it is difficult for the reader to interpret them. Make an effort to simplify the results data so that they can be interoperated more easily.

a.       In 2.2. Bioinformatics the GC content mentioned is 36.80% but in the abstract, it is mentioned as 36.64%, which value is correct?

b.      There are some typological errors that should be taken care of for e.g.: Listeria monocytogenes, Lactococcus, etc. should be in italics.

In 4.7.2. De novo assembly - Illumina- In materials and methods- Authors need to describe in detail.

Some of the references are too old for this research,  update the reference
